# Aerodynamic Characterization of the 516 Arouca Pedestrian Suspension Bridge over the Paiva River

**Fernando Marques da Silva**

National Laboratory for Civil Engineering (LNEC), Av. do Brazil 101, 1700-066 Lisbon, Portugal; fms@lnec.pt

**Abstract:** Given the 516 Arouca pedestrian suspension bridge's design and characteristics, the owner, a municipality, required a set of tests in order to evaluate its aerodynamic characteristics and dynamic response, aiming at both structural safety and user comfort. Wind tunnel tests were performed over a sectional scaled model to obtain the static aerodynamic coefficients and dynamic response. The tests were carried out on different bridge configurations—a deck with people and a deck with an arch for secondary cables (connecting each suspension point to the catenary on the opposite side of the deck)— for the static coefficients. For the dynamic response, only the deck alone was tested. A major challenge had to be overcome, as the main displacement mode belonged to a swing movement, to assemble a wind tunnel setting, requiring a suspension system allowing wind displacements. A persistent trend of small amplitude displacements was identified, influencing user comfort and contributing to the installation of the secondary cables, but no aerodynamic instabilities were identified.

**Keywords:** suspended bridges; wind loading; wind tunnel





## 1. Introduction

Suspension, suspended or cable-stayed pedestrian bridges play a prominent role in reducing distances in remote areas for people to move around, while others serve urban mobility solutions or touristic purposes. Long-span bridges of this kind are of a limited number around the world, most of them being of the suspended type (with no tall end pillars to raise the catenary [1] and usually below 200 m length). However, there are examples of longer bridges up to 494 m (Charles Kuonen in the Swiss alps—85 m of maximum height above ground [2]). These kinds of bridges are also equipped with horizontal catenaries at deck level to prevent excessive swing displacements (Figure 1). Suspension pedestrian bridges also usually use a stiff and straight deck (no sag).

Only very few works on these types of bridges can be found in the literature as the large majority of suspended bridges are for road traffic. Since the Takoma Narrows accident (1940), such bridges are usually submitted to wind tunnel tests worldwide, for example, the ones referred to in [3–7].

The general rules of design for suspended pedestrian bridges are addressed in [1,8], and specific wind actions are addressed in [9,10] for cable-stayed bridges and suspension bridges with solid decks, respectively, or in [11] for a double-deck bridge with porous guards but solid beams and decks.

The bridge under analysis is of the suspension type and is 516 m long—which makes it the longest of its kind—with a 1.2 m-wide deck with a remarkable sag running 175 m above the Paiva river in the North of Portugal. Having no deck level catenaries (usually seen in these kind of bridges, as in [2,12], for example, Figure 1) makes it unique, and having such characteristics makes it very flexible, with a swinging displacement first-mode of a very low frequency, which reinforces the need for wind tunnel testing.

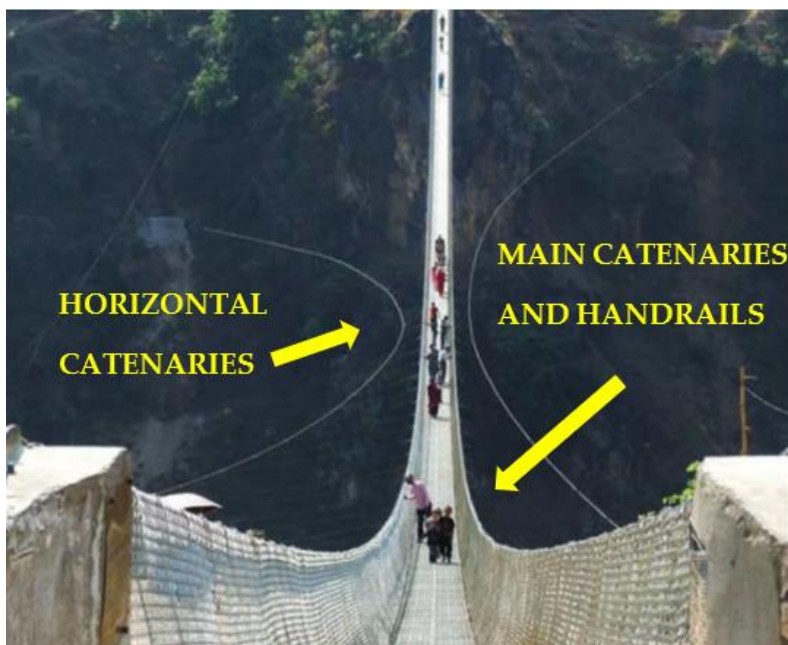

**Figure 1.** The Kushuma Gyadi (Nepal) suspended bridge (photo by Basu Dahal, [12]).

The evaluation of the bridge's aerodynamics became an obvious need leading to a set of wind tunnel tests being performed over a scaled sectional model to obtain static aerodynamic coefficients—drag, lift, and rolling moment—and the dynamic response to determine if any instability may occur. The tests were carried out on different bridge configurations—a deck; a deck with people; and a deck with an arch for secondary cables for the static coefficients (extra tests were performed with no floor mesh and with an opaque floor)–for upward or downward winds. For the dynamic response, only the deck alone was tested, the swing mode requiring a specifically designed suspension for the model as the displacement was along the wing and was not bending or torsional. The wind tunnel dynamic tests aimed to ensure that no aeroelastic instabilities—divergence, galloping, flutter, or vortex shedding—were able to occur up to a determined wind velocity.

The presented work starts with a brief description of the structure, followed by the theoretical basis of the model's testing, the model description, the test facility, and the tests' assembly. The obtained results and conclusions finish the paper.

## 2. Analysed Structure

The site shows a complex orography where the Paiva river flows on an upwind, deep-winding gorge, suggesting that the aerodynamic attack angles may show small departures from zero, mainly downwards from NW and upwards from SE.

The suspension bridge is supported by two reinforced concrete pillars which are 35.5 m tall at each end on the valley sides. The bridge span is 516.5 m, and it is suspended through a system of catenaries and hangers. The distance between the main cables increases steadily from the bridge midpoint to the pillars. The total length of the deck board is 507.6 m, and it is suspended from the main cables by hangers showing a maximum sag of 23.4 m (Figure 2). The geometry of the pillars and the main cable (catenary) is shown in Figure 3.

The deck board consists of 127 metal modules, each one 4016 mm long with a maximum width of 1260 mm. The modules are connected at their upper ends on the same node together with the hangers (Figure 3). The structure modules consist of grade S275 UPN steel profiles and an angle bracket. The lugs connecting to the hangers are located at the outermost point of the frames, the connection being made by an M20 bolt and a single nut. The floor consists of a metallic mesh 1.20 m wide (inner width), and the side guards between the handrail and the floor follow the shape of the frames with a grid. Both

elements have a high porosity (>70%). The geometry of one module with hangers is shown in Figure 3.

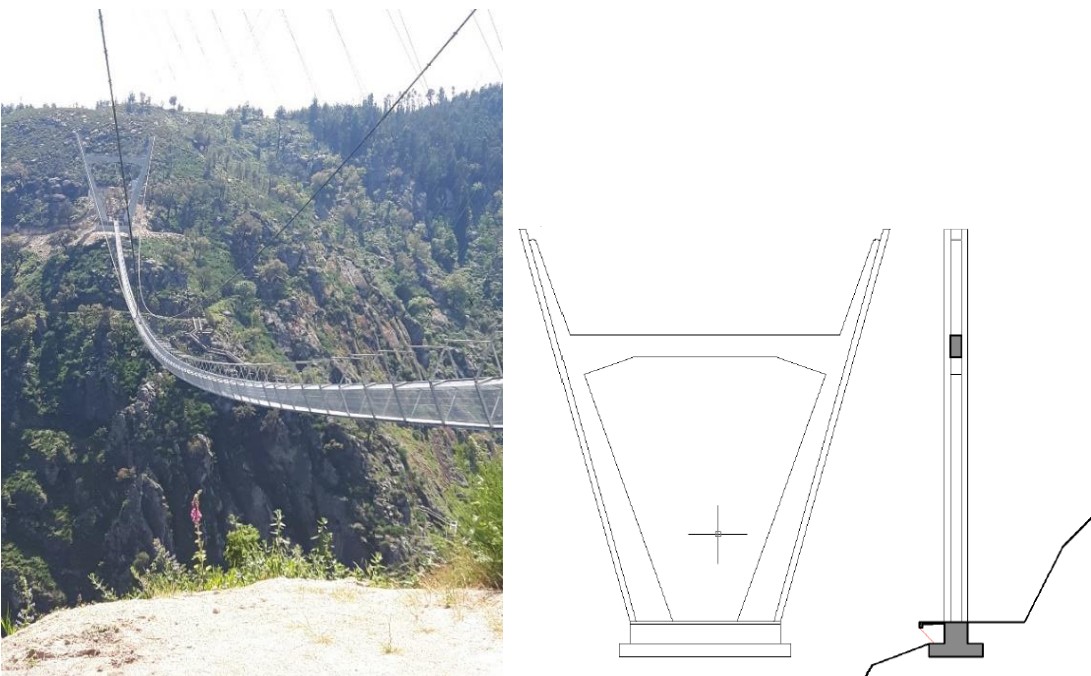

**Figure 2.** The 516 Arouca bridge and pillars geometry.

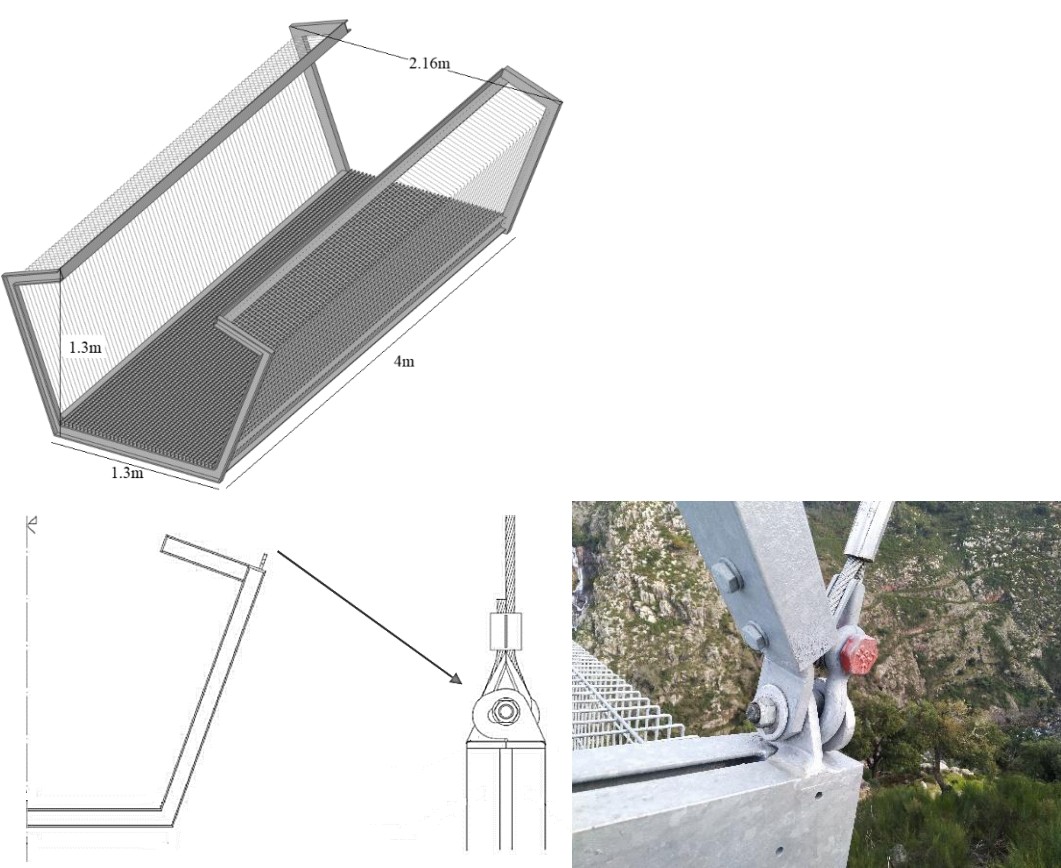

**Figure 3.** Geometry of one module of the bridge and the connection point to the hangers [13].

### 3. Methodology, Model and Test Facility

*3.1. Similitude*

The similitude theory establishes proportional ratios and the scales among physical entities, characterizing the prototype and its behaviour, and the values of homologous physical quantities on the reduced model are also established. For dynamic issues ruled by fundamental dimensions such as length (*l*), mass (*m*), time (*t*), or frequency (*f*), those scales are, respectively, $\lambda$, $\mu$, $\tau$ or, $\varphi$ defined by,

$$\lambda = \frac{l_m}{l_p}; \ \mu = \frac{m_m}{m_p}; \ \tau = \frac{t_m}{t_p} = \frac{f_p}{f_m} = \varphi^{-1}, \tag{1}$$

where indices *p* and *m* refer, respectively, to the prototype and the model. Related entities may be obtained directly from Equation (1) keeping the dimensional homogeneity as the following for force and velocity:

$$\phi = \frac{F_m}{F_p} = \mu\tau\lambda^{-2}; \ \xi = \frac{U_m}{U_p} = \lambda\varphi. \tag{2}$$

When considering the wind structure's interaction with the model characteristics, the flow has to also be taken into account, meaning that the flows around the prototype and the model have to be similar. The ratio between the flow inertia and viscous forces, and the Reynolds number, Re, should be kept constant. However, due to the small dimensions of the model's structural parts of interest—the deck beams and handrails—this is not possible to achieve. Nonetheless, as those parts show sharp edges (meaning clear flow separation points), the flow distortion and resulting pressure distributions are negligible for Re > $10^4$ [14], and the aerodynamic coefficients may be assumed as the Reynolds independent.

The atmospheric wind is naturally turbulent (more so in complex orography) so the tests should be simulated on an appropriate scale to the turbulence spectrum, the integral scale length, $L_u^x$, and the cross and vertical velocity correlations defining the conceptual eddy. However, as low turbulence flows determine heavier conditions for aerodynamic instabilities [15–20], wind tunnel tests performed under low turbulence flows are accepted as contributing to safety. Those kinds of tests are conserved for strong winds such as those with $L_u^x = C\,h^m \approx 290$ m, where C and m are the roughness dependent [18] ($z_0 \approx$ 0.05 m) and h is the deck height, which is much larger than the deck's cross dimension. The characteristic eddy fully wraps the deck for a long enough time (around 20 s for a 50 km/h (14 m/s) wind speed)—turning reasonably to consider an appropriate average velocity and a quasi-static analysis—and the small eddies' contribution may be neglected; the same reasoning applies to the vertical eddy's dimensions, $L_u^y \approx 0.2L_u^x \approx 60$ m, and the cross dimension, $L_u^z = 6\,h^{0,5} \approx 80$ m [18], is much smaller than the bridge length, leading to reduced loads on the prototype compared to the model (the cross correlation becomes negative after a given distance so, for certain modes that may be excited due to asymmetric loading, turbulence effects may lead to a response increase). Recent work [21] refers to the vanishing of vortex-induced vibration (VIV) for turbulent flow as well as the significant reduction of torsional vibration with porous railings.

Turbulence may also affect the static aerodynamic coefficients as it may change the shape of the flow stream lines that "mark" flow separation, and so drag, as the base pressure values also change. For the present case, where the elements contributing to drag are long (crosswind direction) and shallow and have sharp edges, they will go on the way of reducing drag [22,23]. Wind tunnel 3D dynamic tests performed on complete, very long suspended bridge models identified a possible snake-like opaque deck motion originating from gusts [24].

As it concerns the angle of attack, $\alpha$ (Figure 6), increasing the values of the average wind velocity causes the magnitude of the vertical component to diminish so strongly that winds are closely horizontal with $\alpha > \pm 5°$ for average velocities of 80 km/h or $\alpha > \pm 3°$

for average velocities of 120 km/h [22]. For the present case, the orography may induce vertical components, making the wind downwards for northern winds or upwards for southern winds, although with small angles.

The prototype–model behaviour similitude requires the equality of all no-dimensional parameters to be obtained from the relevant physical quantities, so in addition to the geometric scale, it is required that the dynamic behaviour in the model follows the same laws that the prototype does (even if somehow simplified). The use of a wind tunnel to study the structural behaviour of a suspension bridge under the wind load, when inertial forces may play an important role, determines a unitary value for the fluid density and gravity scales leading to $\mu = \lambda^3$ and $\tau = \lambda^{1/2}$. When mass forces are important (as in the present case), the so-called Froude similitude means that only one dimension is arbitrary: the length scale, $\lambda$, due to the wind tunnel test chamber dimensions.

### 3.2. Model

The model scale ($\lambda = 1:4$) was determined by attending to the following:

- The need for enough definition for the deck and handrail beam shapes;
- The need to avoid the effects associated with a low Reynolds number;
- The need to a reproduce a deck module (4 m), roughly 8% of the full length;
- Due to its large porosity, it uses 4% of the wind tunnel cross section so blockage is not an issue;
- The Reynolds number, which was evaluated as Re > $10^5$, using the smallest beam dimension, for the full range of test wind velocities.

The model reproduced two half-deck modules with its U-shaped side, cross beams, floor mesh, double (face-to-face) U-shaped support beams, L-shaped handrails, and guard mesh (Figure 4). Both meshes reproduced the respective porosities. Its total length was 1.0 m, 325 mm wide for the deck and a maximum of 540 mm for the guard, and it was 325 mm high.

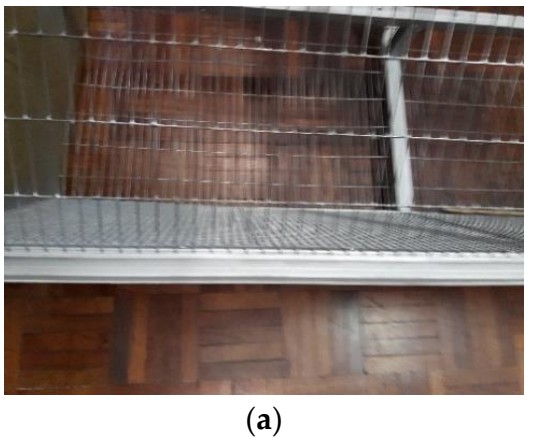
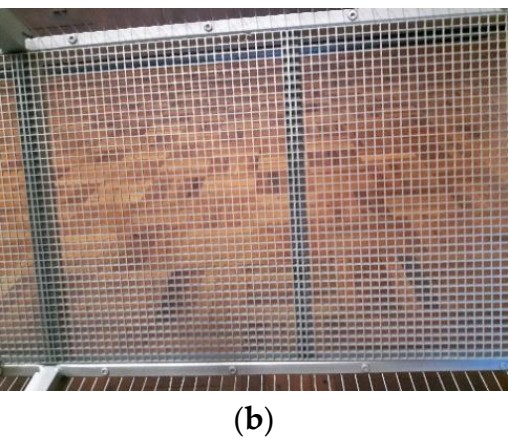

(**a**)    (**b**)

**Figure 4.** The model's: (**a**) deck beam and guard mesh, and (**b**) deck mesh.

In order to keep a 2D flow, avoiding 3D effects around the model tops, thin flat end plates with rounded corners were installed (Figure 5). Connecting to the end plates' sets of supporting external horizontal beams allowed to suspend the model inside the wind tunnel, the attachment point coinciding with the gravity centre.

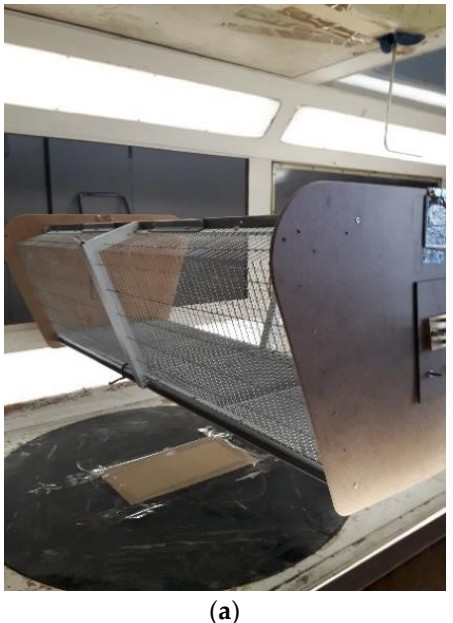 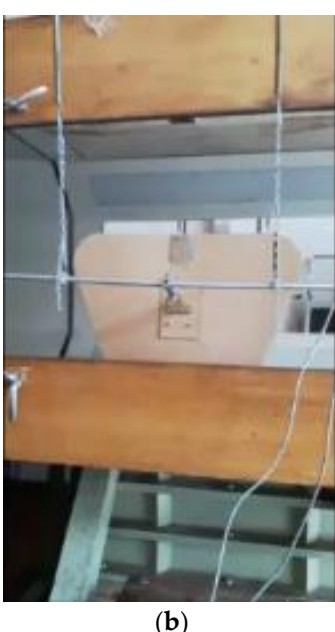

(**a**) (**b**)

**Figure 5.** (**a**) The model installed in the wind tunnel, and (**b**) the model suspension.

### 3.3. Wind Tunnel

The tests were performed in a WT at the Center of Seismic Engineering and Structural Dynamics (NESDE) of Department of Structures (DE) in National Civil Engineering Laboratory (LNEC). The used WT was of the aeronautical type, and had a closed circuit with one fan controlled by a frequency controller, allowing the wind velocity to go up to circa 45 m/s, [24]. The test chamber was $1.0 \times 1.2 \times 3.0$ m$^3$ (the cross section being 1.20 m$^2$) and had a uniform flow with a low turbulence (<1%) [25]. All tests were performed under an open test chamber to avoid blockage.

The wind velocity speed was evaluated from the flow dynamic pressure using a 5 mm Pitot-Prandtl tube connected to a Betz-type precision micro manometer. The forces on the model were measured using six-load cells from Aihasd (5 kg max.)—four for lift and two for drag—including full-bridge strain gauges. The cells' calibration showed a linear force signal behaviour. Calibration and test data were collected by a HBM SPIDER 8 unit.

Dynamic test data were sampled at 100 Hz frequency for a test duration of 25–30 s.

### 3.4. Static Tests

The model was suspended by four sets of chain-spring-load cells (for vertical forces—lift) connecting the horizontal beams to fixed points at the WT's roof level. Drag was measured by a pair of load cells (one at each side) connected to fixed positions on the WT structure and, via INVAR wires, to the model's external horizontal beam.

The aerodynamic angle of attack, $\alpha$ (Figure 6), was obtained by changing the suspension chain's length. A positive $\alpha$ means an upward flow and vice-versa, representing the possible orographic influence [23].

The model was free to move vertically and in rotation but not downwind, the lift forces being evaluated by adding the four vertical load cells measurements, and the moments being evaluated from the difference in the upwind and downwind vertical load cells. Drag was evaluated by adding the two horizontal load cells' measurements.

The aerodynamic forces and moments were then evaluated for a set of wind velocities up to Um = 12 m/s, and the respective aerodynamic coefficients, $C_L$, $C_D$ e $C_M$, lift, drag, and moment, respectively, were evaluated for each tested angle of attack according to

Equation (3) via a linear regression applied to the pairs of F(U)—qA. See Figure 7, where an example of $\alpha = 0°$ is shown, where $q = \frac{1}{2} \rho U^2$, and A is the reference area.

$$C_L = \frac{L}{\frac{1}{2}\rho U^2 d}; \; C_D = \frac{D}{\frac{1}{2}\rho U^2 d}; \; C_M = \frac{M}{\frac{1}{2}\rho U^2 dx}, \tag{3}$$

where $\rho$ is the air density, U is the wind velocity, d = 0.301 m²/m is the sum of the exposed solid beams' widths—deck beam + twice handrail beams + twice frame beams—and x is the distance from the gravity centre to the suspension, $d_{GC}$ (=628 mm). Only one deck beam was considered because of the shelter imposed by the deck floor mesh. For the other parts, the distance between them allowed the flow to recover while also loading the downwind ones. A second test was performed with "users", including the silhouettes of a few persons on the deck, with the same length scale, only for horizontal wind (Figure 6).

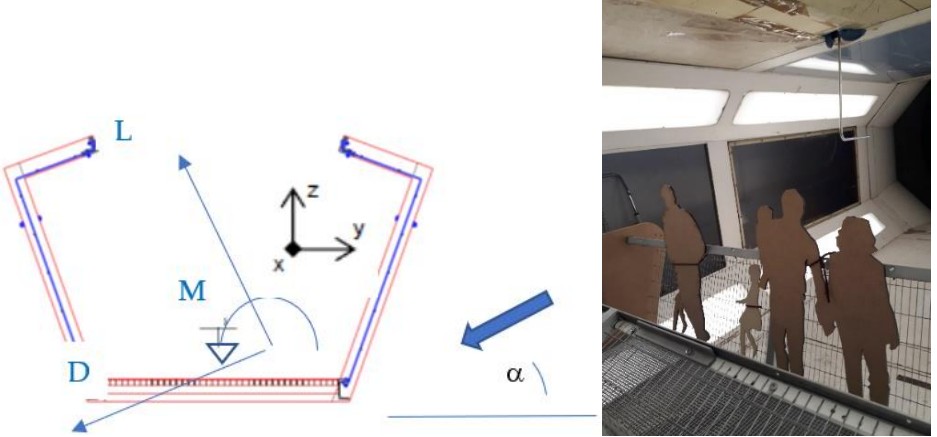

**Figure 6.** Aerodynamic angle of attack, forces (L, D), e moment (M), and silhouettes of bridge users.

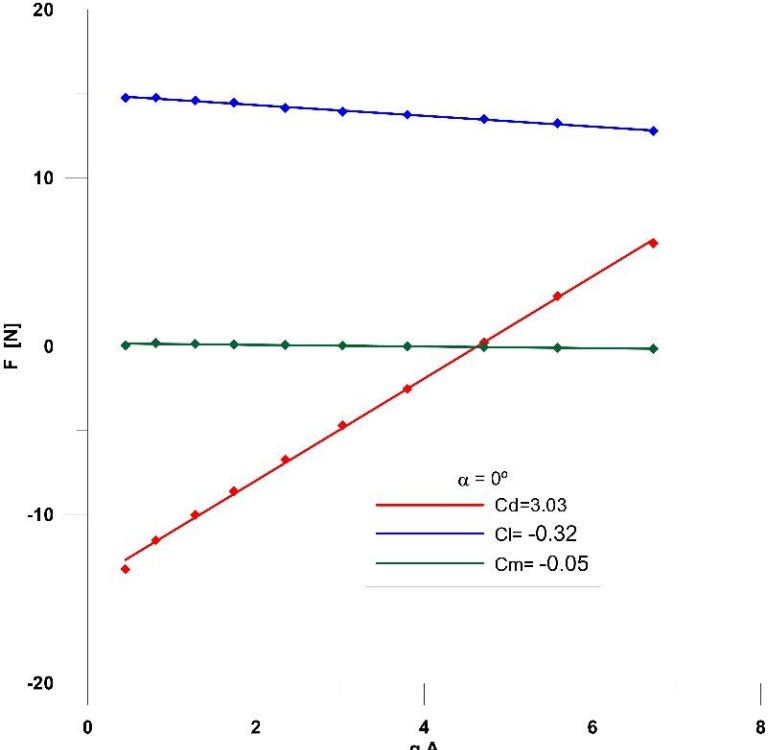

**Figure 7.** Evaluation of aerodynamic coefficients for $\alpha = 0°$.

The added arch, as seen in Figure 8, implied the repetition of the first set of tests to the deck, and as there were 127 arches, new tests were conducted for the arch alone in order to determine its aerodynamic coefficients for varying incidences in the horizontal plane and in the silhouettes of bridge users.

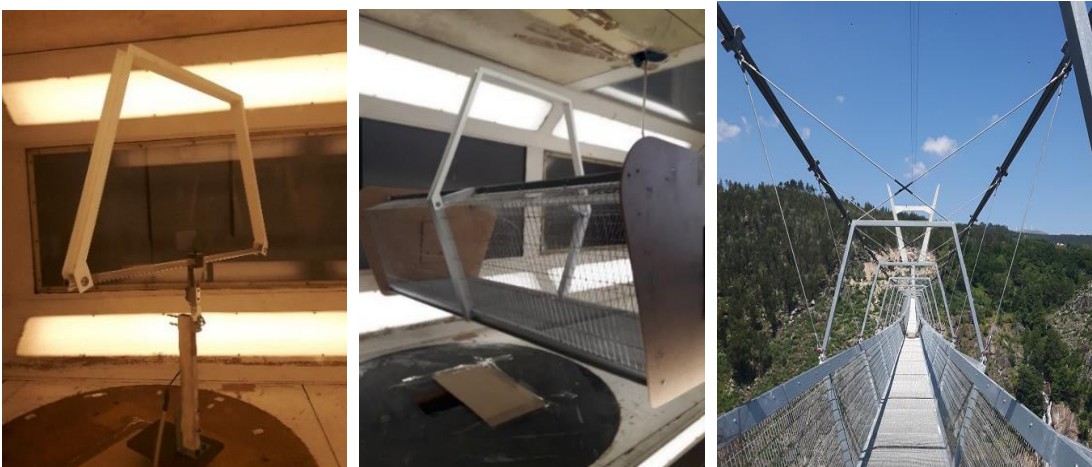

**Figure 8.** Testing the arch, the deck with the arch, and the real system with the secondary cables.

Static tests comprised nine angles of attack for each of the five configurations and for the deck, plus six incidences for the guiding arch, all with wind velocities under ten.

### 3.5. Dynamic Tests

Bridge sectional aerodynamic models are simplified rigid models, but are geometrically detailed enough to be used in wind tunnels to simulate the dynamic behaviour of the prototype and its excitation by the wind in its fundamental bending and torsional modes via an external elastic suspension [18]. As the model shows rigid body displacements, there is no similitude between this and the prototype displacements, so they cannot be directly transposed to the prototype.

This is a long-used technique, (see its use at LNEC [3–6,8]), which allows one to determine the wind velocities that are able to start eventual aerodynamic instabilities, characterized usually by significant deck displacements. Bridge aerodynamic responses focus on vortex shedding (VIV), gallop, divergence, and flutter, but decks are mostly opaque. Slab openings, as in one of the Lisbon bridges [3–5], or porous guards [26] are possible ways to control flutter occurrence. For this particular bridge, there is no slab, so divergence and flutter are not a matter of concern, the beams being the elements that may induce VIV and gallop. Individually, they are too small, but the group may induce a global action.

It was shown [24] that wind loading corresponds to a change in the structure's stiffness and in the damping factor, $\zeta$ (=$\delta/2\pi$), that may be estimated by the logarithmic increase or decrease, $\delta$, of the displacement's amplitude, $a_t$,

$$\delta = \frac{1}{n} \ln\left(\frac{a_t}{a_{t+n}}\right) \tag{4}$$

Instabilities occur if $\zeta i - \zeta a < 0$, where *i* is the mode and *a* refers to the aerodynamics, leading to displacements with a frequency of $\omega = \sqrt{(\omega_i{}^2 - \omega_a{}^2)} \approx \omega_i$ and increasing amplitudes that may, theoretically, grow to infinity.

As the bridge's fundamental mode frequency was very low ($f_0 \approx 0.1$ Hz) and the geometric scale was large (usual bridge sectional models use scales of ~1:70), it became impossible to keep all of the scales referred to above. A time scale was defined, allowing the author to determine the velocity scale ($\xi = \lambda\varphi$, Equation (2)).

Also, as the fundamental mode corresponds to the displacements on the yy' axis (Figure 6), and as the test conditions were unusual, the model suspension assembly was challenging. When swinging, the bridge displacement was in the flow direction, so the suspension springs also had to be positioned along the wind, leading to a change in the suspension system. The problem was overcome by assembling a system (similar to a metronome) using current springs—f = 0.5 Hz, 10 cm below the hinge—and a long enough lever (1.14 m) from the hinge to the suspension point to obtain the appropriate frequency at the wind tunnel mid height where the model is suspended (Figure 9). The load cells were attached to the horizontal deck suspension and the lever was attached to register the "displacement" (via the change in the applied force). However, this system has limitations as the model moves along the wind due to the drag up to the physical limit of the test chamber limiting the maximum test wind velocity.

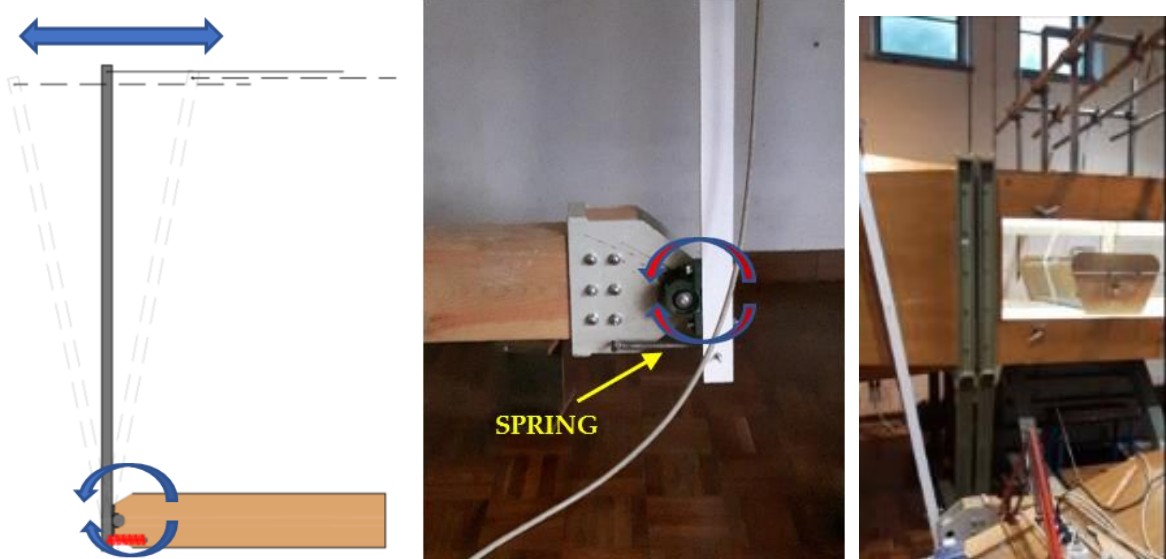

**Figure 9.** System for frequency reduction and model suspension.

A numerical structural model run could not identify any torsional modes (yz plane) coming from the deck, but was only associated to the catenaries movement [27].

Also, antisymmetric vertical bending modes (zz' axis) were identified, but their excitation required non-uniform loading over the structure. For example, the lowest frequency mode showing a central node demanded opposite-direction loads on each half of the bridge, such sceneries being incompatible with the wind loading characteristics.

Nonetheless, the wind tunnel tests evidenced quasi-constant small amplitude torsional shaking leading to the introduction of the secondary cables and the guiding arch (Figure 8) to reduce these rotational movements with consideration of the potential for user discomfort.

Dynamic response tests were then restrained to the horizontal displacements (yy'), the model response being recorded for increasing wind velocities under free displacement conditions and under the previous small-forced displacements.

## 4. Results

### 4.1. Aerodynamic Static Coefficients

The first phase of tests referred to the initial design configuration—deck, guards, and handrail—for angles of attack of $-8° < \alpha < 8°$ (Figure 6), corresponding to downward and upward wind, respectively, followed by the $\alpha = 0°$ incidence with the users' silhouettes—three adults and two children (Figure 6). The second phase included the arch, allowing the author to compare this configuration with the initial configuration. Also, tests on the arch alone for angles of attack of $\Theta = 0°$ (bridge longitudinal direction), 10°, 20°, 30°, 45°,

and 60° (Figure 8), allowed the author to evaluate the added longitudinal loads. Results are plotted in Figure 10 (bridge) and in Figure 11 (other configurations). Figure 12 shows results for the arch drag coefficient for varying incidences along the deck. The first phase's tests results were also used to validate numerical results [28].

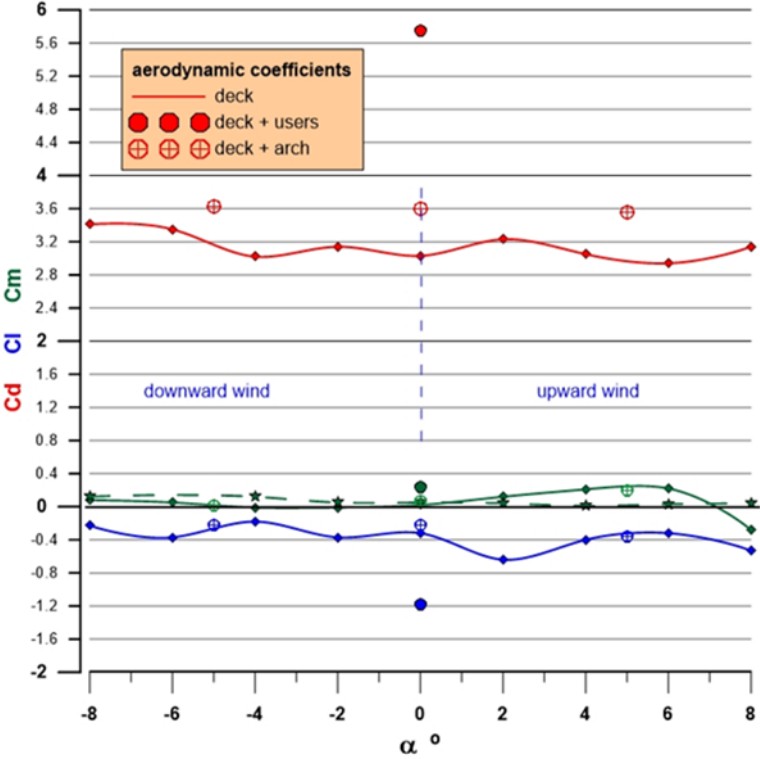

**Figure 10.** Evaluated aerodynamic coefficients.

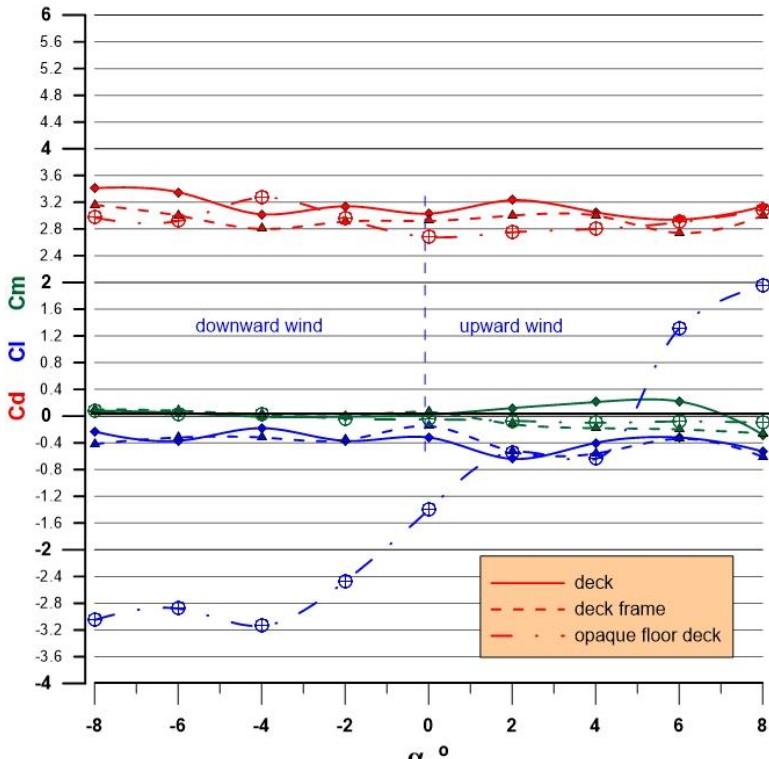

**Figure 11.** Evaluated aerodynamic coefficients for other deck configurations.

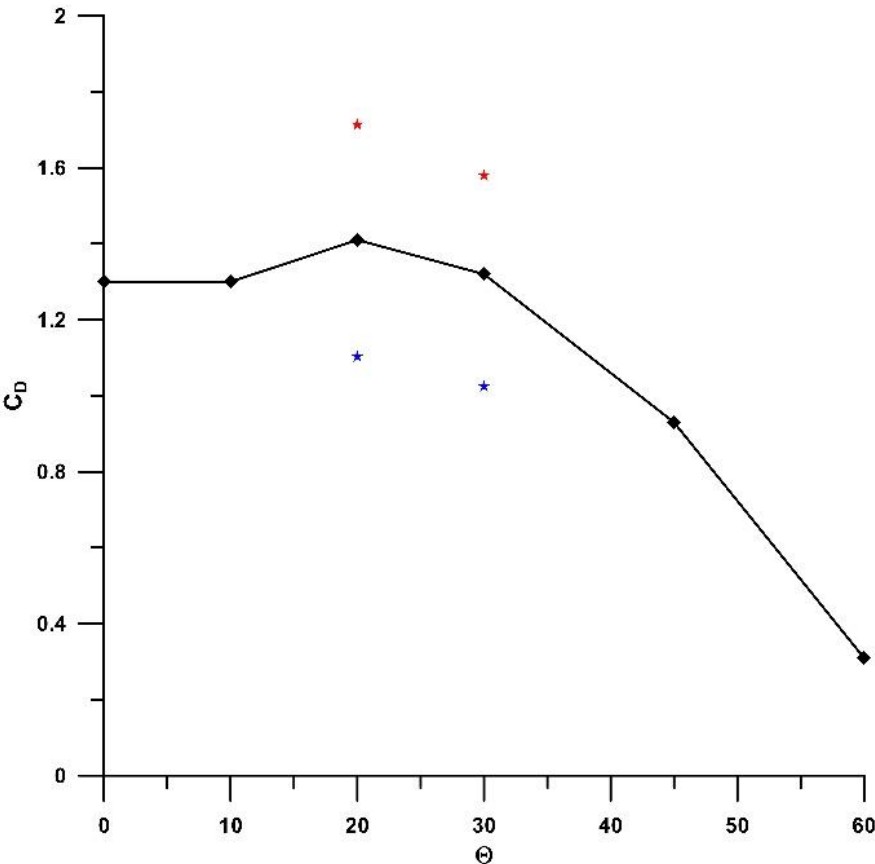

**Figure 12.** Arch drag coefficient for varying incidences.

The opportunity to look into the floor mesh influence and how an opaque floor would change those aerodynamic coefficients was taken, the only significant difference relating, as expected, to the lift evolution getting closer to a flat plate's behaviour or that of other opaque decks [9]. Due to the fact that the model was hung by a flexible system, the question of a possible variation in the angle of attack, when evaluating the static aerodynamic coefficients for a given wind velocity, was raised. As all the tests showed a linear growth in force or moment for a given angle of attack with increasing wind velocity, such a possibility may be discarded, reinforced with the very low values of the moment coefficients.

In Figure 12, the two pairs of coloured points, for $\Theta = 20^{\circ}$ and $\Theta = 40^{\circ}$, above and below the average values, represent the maximum and minimum averages under $f_{vs}$ frequency vibration (see 4.2).

*4.2. Dynamic Response*

The first dynamic evaluation was performed according to the measured forces' time series (Figure 13a) by applying a Fast Fourier Transform (FFT) analysis to check the possibility of vortex shedding (VS) in the arches. The significant amplitudes for higher wind velocities and $\Theta = 20^{\circ}$ and $30^{\circ}$ came from the vortex shedding frequency of fvs $\approx 10$ Hz (Figure 13b), the corresponding Strouhal number (St = fd/U) being St = 0.01. This vibration justifies the arch $C_D$ values' behaviour for those incidences (Figure 12).

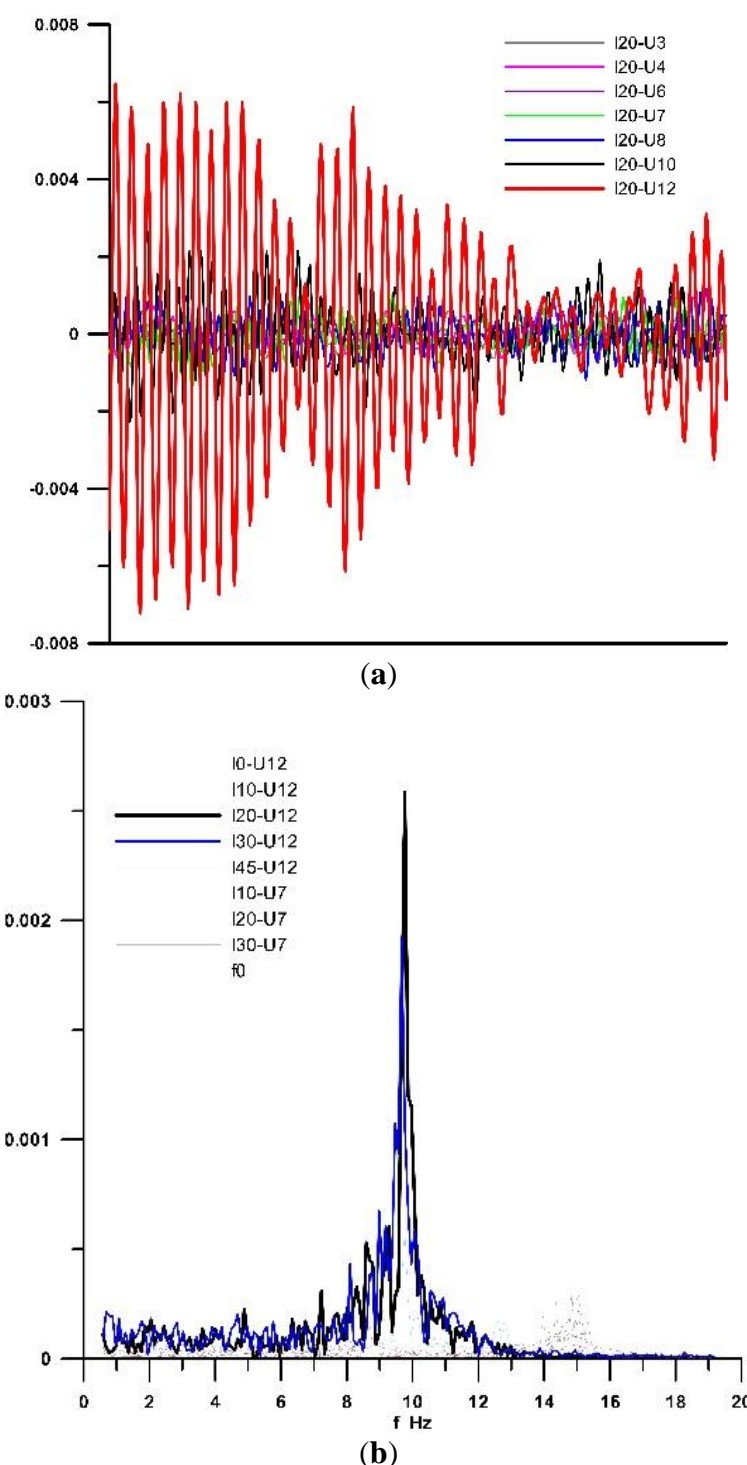

**Figure 13.** Arch force time series and corresponding spectral plot for each pair of -U test condition (**a**,**b**).

There is no evidence of VS excitation for the deck. The deck dynamic response is evaluated by the damping coefficients, $\zeta$. However, the use of logarithmic decay needs the premise of a sinusoidal wave, which does not happen for tests with no external excitation (as shown in Figures 14 and 15). Hence, the presented damping coefficients refer only to situations of previous external excitation on the yy' direction.

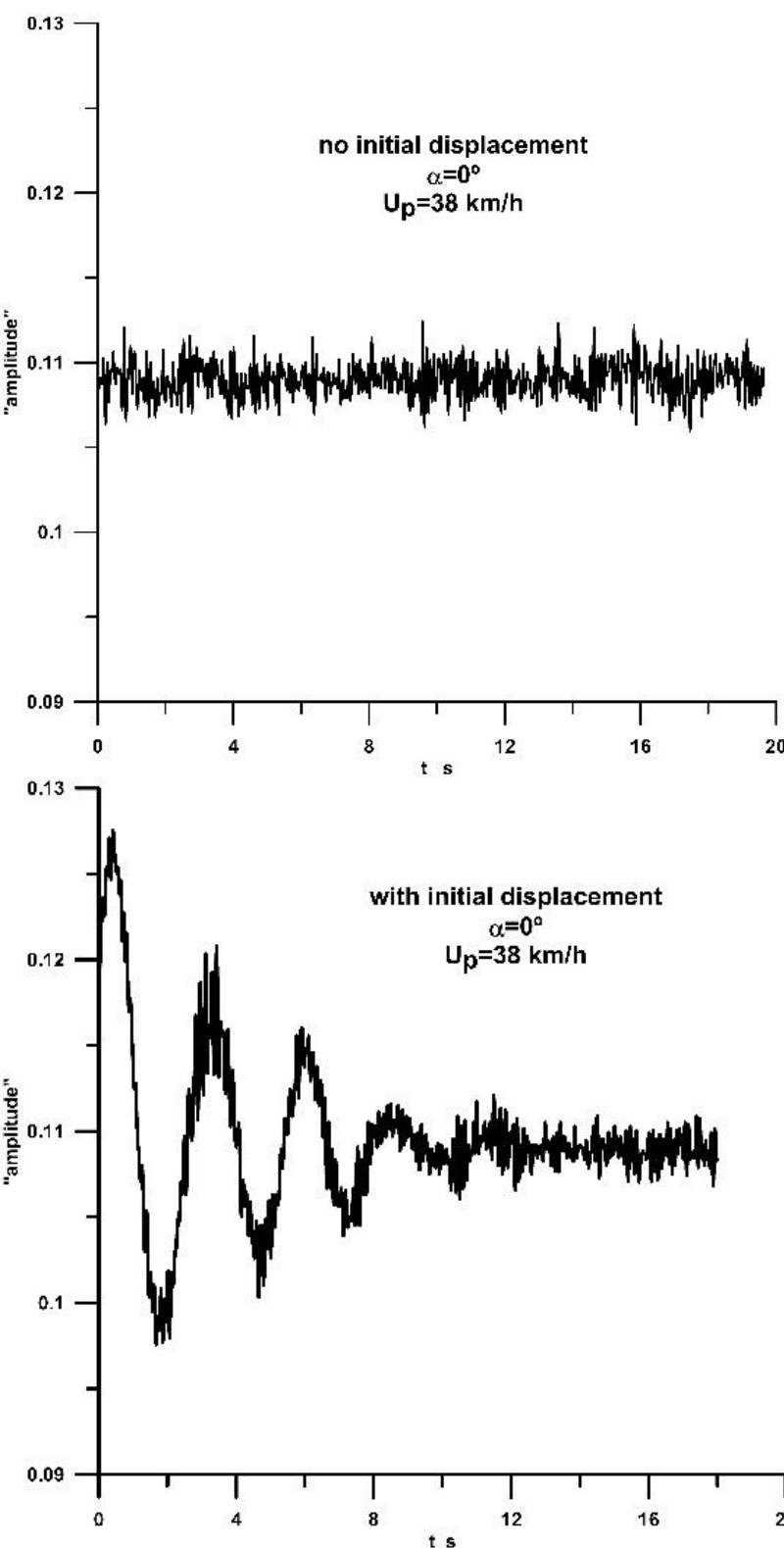

**Figure 14.** Deck torsional movement without and with previous swing displacement.

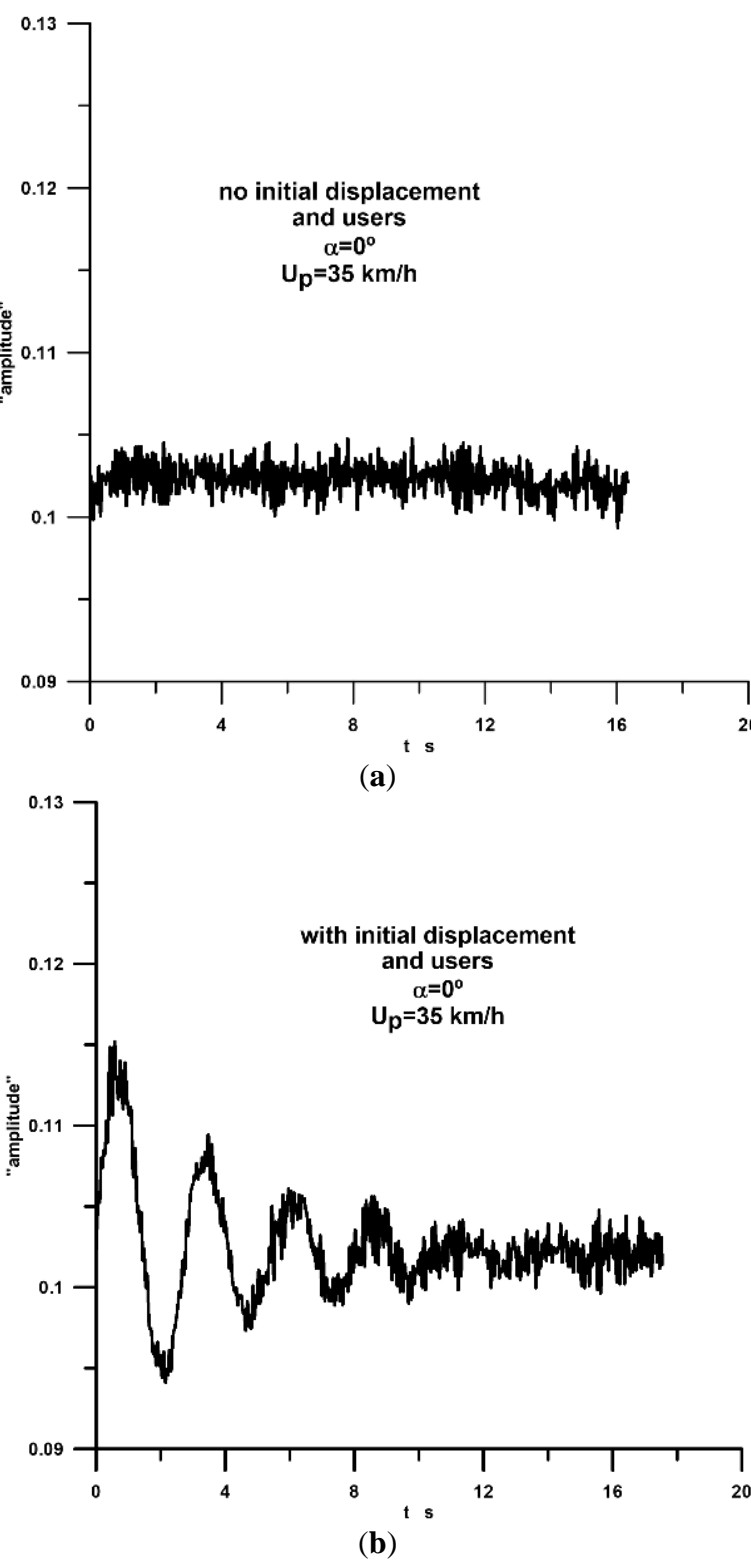

**Figure 15.** Deck torsional movement without and with previous swing displacement with users (**a**,**b**).

The damping coefficient evolution, as a function of both wind velocity and the angle of attack, is shown in Figure 16a, its values being always positive with the initial displacement, drag playing a role in this behaviour.

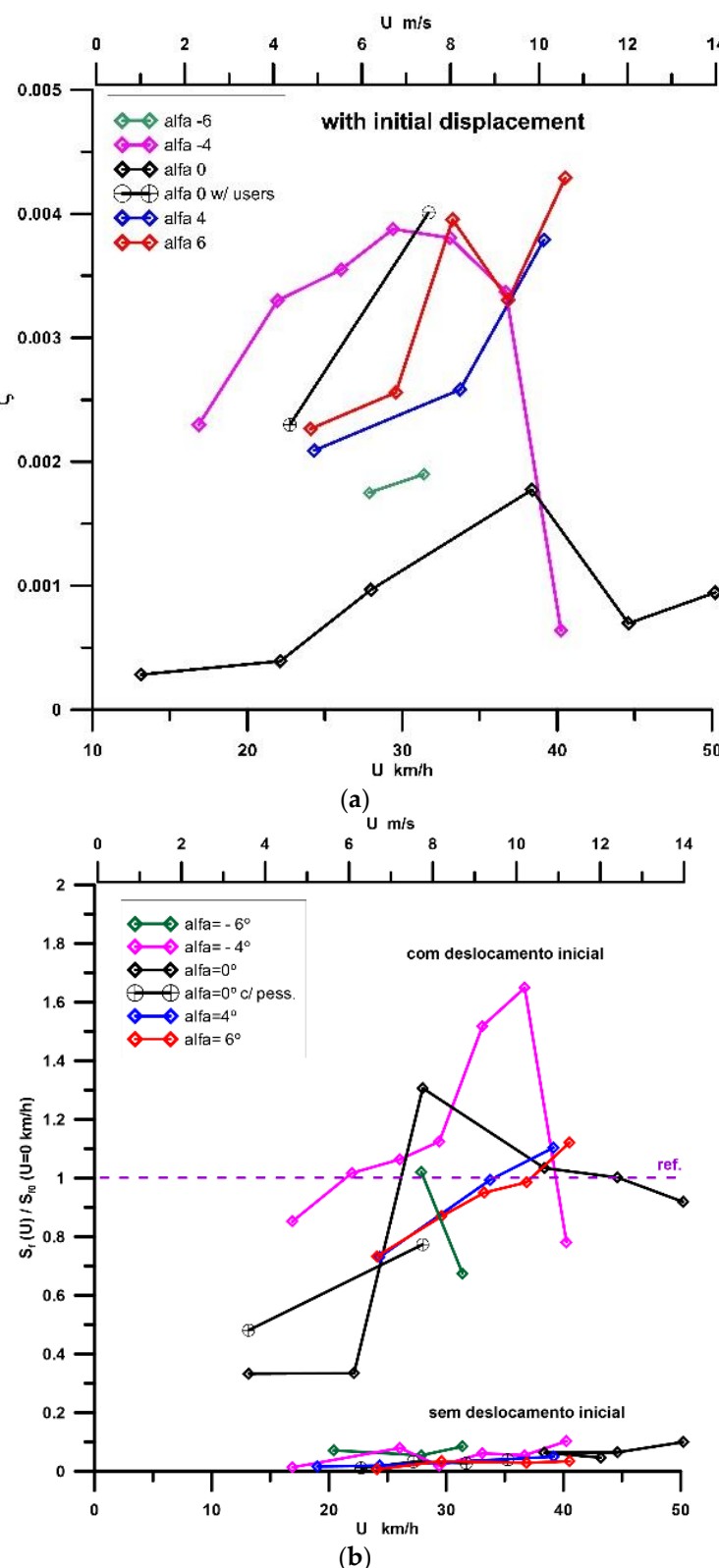

**Figure 16.** (**a**) Damping coefficients for swing displacement; (**b**) (Sf)$_{pk}$ to Sf0 ratios without and with initial displacement.

The vibration relevance is evaluated using the ratio of the spectral component, Sf (from a given frequency, f), to the spectral component, Sf0, of the first mode with no wind (Figure 15b).

Northern downward wind, $\alpha = -4°$, with an initial displacement induced a response with the highest energy content within a range of velocities 20 km/h < Up < 40 km/h. Horizontal winds also showed significant energy content for the same velocity range. With no initial displacement, the energy content of the vibrations was residual.

Figure 17 shows the ratios of the relevant spectral components (Sf > 0.75 Sf pk) to the peak values for each incidence. The indicated wind velocities refer to the prototype values.

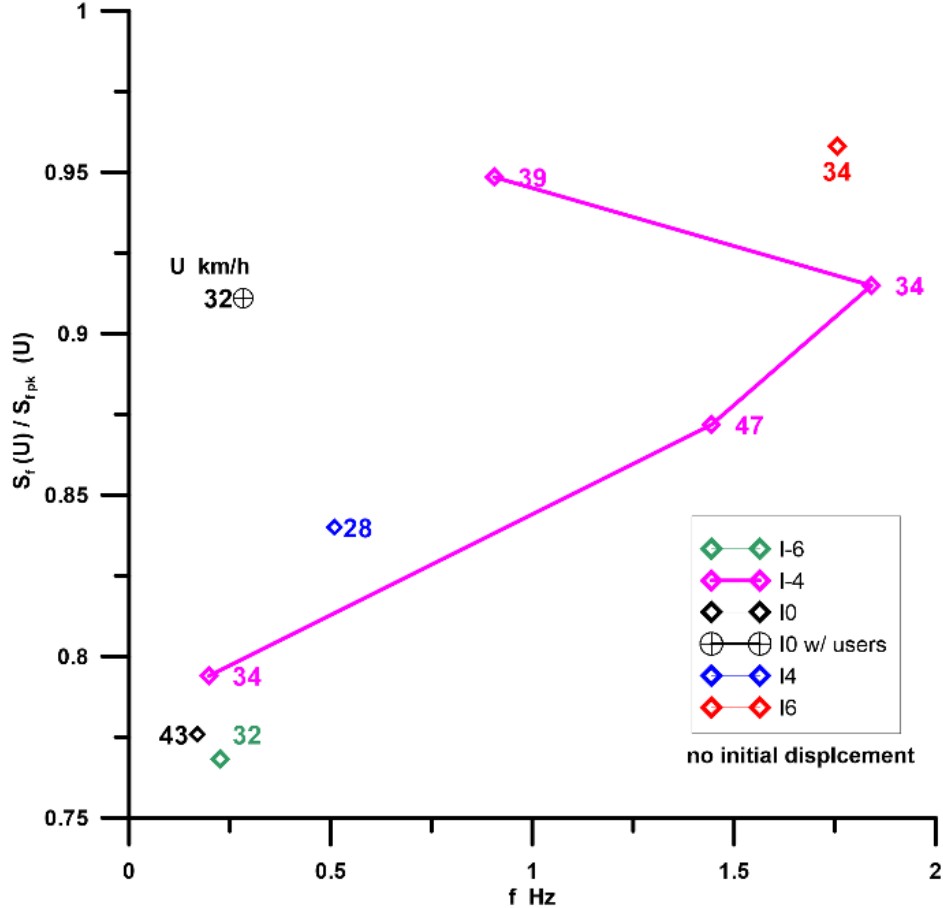

**Figure 17.** Ratios of significant Sf values—Sf > 0.75 Sf $_{pk}$—to (Sf)$_{pk}$.

The spectral peak component corresponds to a frequency that is always close to the first mode and to all of the other significant modes, with the spectral content below the peak occurring for wind velocities above 30 km/h (8 m/s). Within this group, most of the highest spectral components occur for frequencies clearly above the fundamental mode (f0 = 0.116 Hz), and so are unable to excite possible instabilities. The lower frequencies with a significant spectral content are approximately $2f_0$, occurring for $-6° < \alpha < 0°$ (including the presence of users).

## 5. Discussion

An experimental study on the aerodynamic performance of a long-span pedestrian suspension bridge, the 516 Arouca bridge in a hilly location in Portugal, was performed by wind tunnel testing over a deck sectional model. Both static and dynamic tests were performed.

Static tests allowed the author to evaluate the aerodynamic coefficients of drag, lift, and rolling moment, defining the applied wind loads. Different configurations were tested: (i) a deck, handrails, and guards; (ii) a deck with added user silhouettes; (iii) a deck with an added arch for the guidance of secondary cables (added in a project upgrade); (iv) a floorless deck; and (v) an opaque deck.

Table 1 resumes the maximum static aerodynamic coefficients for angles of attack in the range $-2° < \alpha < 2°$. Configurations (iv) and (v) were meant to be used to check the aerodynamic significance of the floor mesh and a "what if" configuration, respectively.

**Table 1.** Maximum aerodynamic coefficients for small angles of attack.

|  | $\mathbf{C_D}$ | $\mathbf{C_L}$ | $\mathbf{C_M}$ |
|---|---|---|---|
| (i) deck | 3.23 | −0.64 | 0.21 |
| (ii) deck + arch | 3.60 | −0.22 | 0.07 |
| (iii) deck + users | 5.75 | −1.18 | 0.23 |
| (iv) deck frame | 2.92 | −0.15 | 0.06 |
| (v) opaque deck | 3.38 | −2.37 | −0.03 |

The presence of the arches introduced bridge longitudinal drag, so additional tests were performed, and average maximum coefficients of $1.3 < C_D < 1.4$ for $0° < \Theta < 30°$ were found. The vibration induced by vortex shedding on the arches increased the range of peak values to $1.0 < C_D < 1.8$ for $20° < \Theta < 30°$. The identified vortex shedding is characterized by a Strouhal number of St = 0.01.

The moment coefficient, $C_M$, trend shows a positive slope for $0° < \alpha < 4°$ (Figure 11), which could indicate a possible torsional divergence problem [16]. Nonetheless, this is not a real problem due to the very low slope value (0.05) and, mainly, because the slope became negative for $\alpha > 4°$, preventing an increase in the angle of attack.

Dynamic tests were restrained to the swing mode displacement as previous numerical structural evaluations showed that torsional displacements were associated to the catenaries only. Also, antisymmetric vertical bending modes (zz' axis) require unrealistic wind loading characteristics. Wind tunnel tests evidenced quasi-constant small amplitude torsional shaking leading to the addition of secondary cables (guided by the arches) to reduce this motion, allowing to minimize discomfort to the users.

The main displacement mode (swinging) with a very low frequency (about 0.1 Hz) required a challenging test installation as the suspension and springs should also have had the direction of the movement. A metronome-like installation was assembled, allowing the reproduction of the swinging movement using common springs, although the displacement amplitude was limited by the test chamber length.

Aerodynamic stability was evaluated by the damping coefficients—all being positive—and, for non-sinusoidal movement, by comparing the most significant spectral energy contents of each test to the peak values.

No aerodynamic instabilities were found, although upward (southern) and horizontal winds showed a significant displacement energy content for velocities above 30 km/h, suggesting that users may feel discomfort.

Additionally, mainly due to the large porosity of the deck and side guards, the structure did not show significant sensitivity to wind, but as drag increased significantly with the presence of people, a maximum wind velocity was defined to allow the bridge crossing to facilitate the users' comfort and feeling of safety.

## 6. Concluding Remarks

A new testing installation was designed to evaluate the dynamic response of a bridge exhibiting a swinging first-mode of a very low frequency.

This installation had to cope with a displacement in the flow direction and a frequency decomposition in order to be able to use regular springs and feasible displacements.

**Funding:** This research received no external funding.

**Data Availability Statement:** Data is unavailable due to privacy issues, although publishing permission was given.

**Conflicts of Interest:** The author declares no conflict of interest.

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
