# Peer review of "Aerodynamic Characterization of the 516 Arouca Pedestrian Suspension Bridge over the Paiva River"

_2674-032X, doi:10.3390/wind3010006_

Round 1

Reviewer 1 Report

This paper would be of interest to civil engineers as well as wider academic community interested in aerodynamics.  It is within the scope of the journal. The title is ok, but the abstract needs work as it does not summarize the contents well. Some key findings should be presented. The structure of paper is good. A good background is given acknowledging appropriate previous work. The problem is well specified.

Only some changes are recommended to improve clarity and remove typos

Rework abstract to add more detail and include more key findings.

Line 21, replace ‘anyway’ with ‘However’

Line 29 make not makes

Line 23 and 29 use correct citation

Line 41 on new page

Figure 1 and 2 could be combined 1a  and 1b

Line 98 VIV not yet defined

Missing symbols throughout needs checking lines 108,122, 161, 170,171, 278

Line 149 states open test chamber but photos in fig 5 seem to be closed test section, some explanation needed

Line 213-215 needs rewriting, unclear

Line 319 table not tale

Author Response

Thank you for you comments that allowed to improve the text.

Reviewer 2 Report

The paper presents a description of wind tunnel tests carried out for a model of the 516 Arouca pedestrian suspension bridge and the results of these tests.

The main criticism is that the article has been prepared very sloppy, which makes it unreadable at times.

Detailed comments:

1. In the introduction, there is no literature review related to the topic of the work at all.

2. No explanations for the abbreviations used or explanations are much further in the text than the first use of the abbreviation, e.g.: line 98: VIV, line 127: WT, line 227: FFT analysis.

3. Many symbols are missing in the text, eg.: line 108, 112, 161, 170 and furtther.

4. Many symbols are written once with subscript and other times in normal font, eg.: line 273: fvs, line 274: CD.

5. Incomprehensible expressions, eg.: line 180: "forces e moments".

6. The title of Figure 7 is completely inadequate to the content.

Author Response

Thank you for the comments that allowed to improve the text

Reviewer 3 Report

- P1; L13: add a little bit about your findings.

- P1; L25: It is recommended to add a figure illustrating the different components of the suspension bridge

- P1; L40: This reviewer recommends adding a paragraph at the end of the introduction section to present the manuscript layout.

- Figure 3: Please add illustrations to the figure and the dimensions are not precise.

- P4; L98: what is the meaning of "VIV"?

- P4; L108: de??

- I guess there is a problem with text formatting and equations, for example, P4; L122

- P4; L127: meaning of "WT"??

- P5; L138: de ??

- P5; L143: The first letter is usually capitalized in each word for the terminologies

- P6; L161&L162: There are many locations with missing symbols.

- Reference for equation 3.

- Is there any warping for text at the vertical axis of fig. 11 and 12?

- Improve the quality of fig. 13, 14, 15, 16, and 17.

- P11; L271- 277: meaning of "FFT & VS"??

Author Response

(The authors gave the same response as above.)

Reviewer 4 Report

This paper discusses the aerodynamic characteristics of a pedestrian suspension bridge in wind tunnel testing. Some comments are provided below:

Line 16:The writing in the introduction is not presented clearly and logically, and needs some revision for clearer presentation. The authors are urged to start with a clear description of the problem statement. Subsequently, a summary of conclusions from past research should be accompanied with the scope of the present research (e.g., which gaps are to be filled and why that is important) and a clear description of the methodology followed.

Line 91: Please standardize the format of references.

Line 135: Figure 4, 5 and 7 are vague, not depicting the significant components of experimental setup, thus, requires clarity and content and sketching quality.

Line 156: The authors should provide more details regarding the wind tunnel tests, at least including how many sets of tests and test duration.

Line 161: Parts of this sentence are missing.

Line 190: For dynamic tests, the authors should provide more details regarding the wind tunnel tests, at least including the main parameter of the tested sectional model system.

Line 278: Parts of this sentence are missing.

Author Response

(The authors gave the same response as above.)

Round 2

Reviewer 2 Report

The most important corrections have been incorporated in the text. In my opinion, the paper can be published in present form.

Author Response

The most important corrections have been incorporated in the text. In my opinion, the paper can be published in present form.

Thank you very much for your opinion.

Reviewer 4 Report

1)     The answer to my previous comments is not satisfied. The previous research has not been clearly reviewed in the introduction part. The problem is not defined logically. The experiment method is not clearly introduced.

2)     The figure quality in this manuscript is low. The type size and line width are not proper.

Most of the figures should be redesigned to satisfy the requirement of academic paper.

3)     No conclusion is given. There should be general conclusions which can be referred by other researchers.

Author Response

Thank you for your revision. The following updates were introduced.

1)     The answer to my previous comments is not satisfied. The previous research has not been clearly reviewed in the introduction part lns 31-35. The problem is not defined logically lns 41-43. The experiment method is not clearly introduced lns 53-54.

2)     The figure quality in this manuscript is low. The type size and line width are not proper. Figures size were increased

Most of the figures should be redesigned to satisfy the requirement of academic paper.

3)     No conclusion is given. There should be general conclusions which can be referred by other researchers. A final paragraph was added to Discussion.

Round 3

Reviewer 4 Report

1) The answer to my previous comments is not satisfied. The figure quality in this manuscript is very low and not professional.

Most of the figures should be redesigned to satisfy the requirement of academic paper.

2) The analysis of the measurement data is not innovative nor deep.

3) The  conclusion is not proper. There is no numberred conclusion that summarizes the main new findings of this research. 

Author Response

The figure quality in this manuscript is very low and not professional. Most of the figures should be redesigned to satisfy the requirement of academic paper." - I am not an editing professional and I can not do better with the software I have always used.

"The analysis of the measurement data is not innovative nor deep" - Evaluating the aerodynamic behaviour of a bridge is usually not an innovative procedure. Anyway this particular bridge required an innovative testing procedure as is clearly stated in the text. Plus this was the main reason to write the paper.

"The  conclusion is not proper. There is no numberred conclusion that summarizes the main new findings of this research" - I believe that the conclusions are clear although not surprising or innovative.